# Human-like Biofilm Models to Study the Activity of Antifungals Against *Aspergillus fumigatus*

**DOI:** 10.3390/microorganisms13092040

**Published:** 2025-08-31

**Authors:** Dan-Tiberiu Furnica, Julia Falkenstein, Silke Dittmer, Joerg Steinmann, Peter-Michael Rath, Lisa Kirchhoff

**Affiliations:** 1Institute of Medical Microbiology, University Hospital Essen, University of Duisburg-Essen, 45122 Essen, Germanysilke.dittmer@uk-essen.de (S.D.); joerg.steinmann@klinikum-nuernberg.de (J.S.); peter-michael.rath@uk-essen.de (P.-M.R.); lisa.kirchhoff@uk-essen.de (L.K.); 2Institute of Clinical Hygiene, Medical Microbiology and Infectiology, Klinikum Nürnberg, Paracelsus Medical University, 90419 Nuremberg, Germany

**Keywords:** *Aspergillus*, fungal biofilm, ex vivo, precision cut lung slices (PCLS)

## Abstract

*Aspergillus fumigatus* is an opportunistic filamentous fungus that primarily affects the respiratory tract of the human body. Depending on its host’s immune response, the pathogen can cause invasive pulmonary aspergillosis (IPA). Biofilm formation by *A. fumigatus* increases virulence and resistance against antifungals and immune response and is one important factor in IPA development. Here, two human-like models, precision cut lung slices (PCLS) and a biofilm co-culture model, have been developed to test the anti-biofilm activity of voriconazole, amphotericin B, as well as luliconazole against *A. fumigatus*. In both assays, metabolically active *A. fumigatus* biofilms were examined at different biofilm developmental stages using an XTT assay. A decrease in the metabolic activity of the fungal biofilms was detected for each of the tested agents in both assays. Significant anti-biofilm effects exist against early-stage biofilm in the co-culture model. In the PCLS assay, amphotericin B showed the strongest inhibition after 24 h. In conclusion, the applied PCLS ex vivo model can be used to study the property and activity of certain antifungal compounds against *Aspergillus* biofilm. With its close resemblance to human conditions, the PCLS model has the potential for improving the current understanding of biofilm treatments in laboratory settings.

## 1. Introduction

*Aspergillus fumigatus* can be considered as one of the most prevalent fungal opportunistic pathogens present in clinics [1]. The World Health Organization recently classified it as a ’critical priority’ pathogen in the fungal priority pathogen list 2022, posing a great risk to public health [2,3].

*A. fumigatus* is a filamentous mold typically found in nature in soil or near compost heaps [4]. The mold replicates through airborne asexual conidia, which are easily dispersed by the wind [5]. Healthy individuals are protected, among others, by macrophages, monocytes and neutrophiles, which stop the germination of conidia into hyphae [6]. Individuals without a correctly functioning immune system, however, are very susceptible to *A. fumigatus* infections and are more likely to develop cases of invasive pulmonary aspergillosis (IPA) [2]. Among the risk groups are patients suffering from hematological disease, chronic lung disease, and solid organ or bone marrow recipients [7]. One factor that makes *A. fumigatus* a serious threat for these patient groups and generally for the public health system is the formation of biofilms. Biofilms contribute towards an increased resistance to external factors such as temperature, pH changes or antifungal compounds [8,9,10]. The formation of an *Aspergillus* biofilm specifically starts with the swelling of the conidial cells within the first 12 h of incubation, followed by surface adhesion mediated by the development of an extracellular matrix (ECM) [11]. The ECM acts as a physical barrier towards antifungal agents, at the same time creating hypoxic microenvironments where the fungus can survive [12]. Due to the drug-resistant nature of *A. fumigatus* biofilms, standard antifungal therapies seem to be becoming less efficient [11]. Furthermore, a discrepancy between in vitro and in vivo *Aspergillus* biofilms could also arise, depending on the specific in vitro cultivation conditions [13]. That is why the investigation of fungal biofilms and the development of novel strategies for an efficient treatment has become more relevant today. Among these strategies, the discovery of novel therapeutic options is also included. In previous publications, we demonstrated an activity of multiple antifungal drugs, including the imidazole luliconazole (LLCZ), both in vitro and in vivo, against the growth and biofilm development of *A. fumigatus* [9,10]. Even though the data is promising, further research is needed considering the already-mentioned discrepancies between in vitro and in vivo as well as planktonic vs. sessile culture results [13].

Only limited in vivo data describing the biofilm formation of *A. fumigatus* has been published to date. A previous study shows significant differences between in vitro and in vivo *A. fumigatus* biofilms, especially in the composition of the ECM. Polysaccharides such as galactosaminogalactan or galactomannan were shown to be more prevalent in vivo [14]. This may have a direct impact on the translation of in vitro to in vivo experiments. The development of fungal biofilm experiments could help close this gap in the current data. Precision cut lung slices (PCLS) are an ex vivo model, increasingly recognized due to its cellular complexity, that may help solve this problem [15]. This model has previously been used in multiple publications to study the planktonic growth of bacteria and the interactions with its host; however, to date, it has never been used to study fungi or fungal biofilms [16,17].

The purpose of this study was to demonstrate the anti-biofilm activity of voriconazole (VCZ) and amphotericin B (AMB) as well as LLCZ during growth conditions similar to those found in vivo. For this purpose, two separate human-like models have been adapted and optimized: a PCLS ex vivo mouse lung model and an in vitro co-culture model with human epithelial lung cells. These models allow experimental analyses of the antifungal drug activity.

## 2. Materials and Methods

### 2.1. Aspergillus fumigatus Strains

A total of 20 *A. fumigatus* strains were used in this study. Species identification was performed using characteristic micro- and macro-morphological criteria and additionally via AsperGenius 1.0 multiplex real-time PCR (PathoNostics, Maastricht, The Netherlands). The majority of the included strains were azole-resistant (N = 17 azole-resistant strains with mutations in the cyp51A gene; N = 3 wild-type strains). Of these, 16 possessed a TR34 mutation, while 1 carried a TR46 mutation. Ten of the isolates were used in the PCLS model while all twenty isolates were used for the co-culture model. All isolates used in this study are listed in Table 1.

### 2.2. Preparation of the Fungal Suspensions

The *A. fumigatus* isolates were first cultivated on Sabouraud agar (Oxoid, Wesel, Germany) for 48 h at 35 °C. The surfaces of the plates were washed with 6–7 mL sterile distilled H_2_O (containing 0.1% Tween20 (Carl Roth, Karlsruhe, Germany) and scraped with cotton swabs to loosen and detach the conidia. The fungal suspensions were collected with a syringe and filtered (10 μm pore size, syringe Filcons, BD, Franklin Lakes, NJ, USA) into sterile 12 mL tubes. The tubes were then centrifuged (3 min, 3000× *g*) and the resulting pellet was washed twice with ice-cold 1x phosphate-buffered saline (PBS) (Sigma-Aldrich, St. Louis, MO, USA). Next, 10 µL of the fungal suspension was diluted into 990 µL PBS, the cells were counted in a haemocytometer (Neubauer, Marienfeld, Germany) and the final inoculum was set to 1 × 10^5^ through dilution with the experiment-appropriate medium. Colony-forming unit (CFU) controls were then performed [10].

### 2.3. Drug Preparation

LLCZ, VCZ and AMB were purchased from Sigma-Aldrich (St. Louis, MO, USA) and diluted with dimethyl sulfoxide (DMSO) (Carl Roth, Karlsruhe, Germany) into stock solutions that were stored at −20 °C. The stock was then diluted to various working concentrations via further dilution in DMSO, followed by a final 1:100 dilution in the assay-specific medium, leaving only a trace concentration of DMSO in the aliquots. VCZ and AMB were diluted (in DMSO and then in the assay-specific medium) to concentrations of 1 mg/L which correspond to the EUCAST recommended breakpoints for *A. fumigatus* [18]. For LLCZ there were no known breakpoints, so several concentrations (0.002–0.06 mg/L) corresponding to the previously determined [10] strain-specific MICs were used (see Table 1). A final DMSO concentration of 1% remained in the antifungal drug aliquots.

### 2.4. Cultivation of A549 Cells

The cells chosen for co-culture experiments with *A. fumigatus* biofilm were A549 human lung carcinoma epithelial cells, kindly donated by the Department for Molecular Biology, University Duisburg-Essen (Working Group Becker-Flegler). A549 cells were cultured in 75 cm^2^ polystyrene cell culture flasks (Greiner, Bio-One, Frickenhausen, Germany) using DMEM (12320032; Fisher Scientific, Gibco, Schwerte, Germany) supplemented with 10% fetal calf serum (FCS; Fisher Scientific, Gibco) and 1% penicillin–streptomycin (Fisher Scientific, Gibco). The cells were grown until confluence was reached and confirmed through microscopy (90% of the flasks bottom was covered in cells). The supernatant was then discarded, and the flask was rinsed twice with 1 mL of a 1 × Trypsin-EDTA solution (Sigma-Aldrich, St. Louis, MO, USA). The cells were then incubated for 5 min at 37 °C and 5% CO_2_ and re-suspended in 10–12 mL DMEM. The final working concentration for the cells was 1 × 10^5^ cells/mL, which was confirmed with a fluidlab R-300 cell counter (Anvajo, Dresden, Germany) [19].

### 2.5. Biofilm Co-Culture with A549 Cells

The following assay was used to investigates the effect of the antifungal compounds against immature (4 h), maturing (12 h) and mature (24 h) *A. fumigatus* biofilm formed on an A549 cell line. Fungal suspensions were prepared as described in Section 2.2 and set to an inoculum of 1 × 10^5^ cells/mL. The cells and the conidial suspensions were incubated together, either in a 96-well flat-bottomed microtiter plate when early-stage 4 h old biofilm was assessed (100 µL cells and 100 µL conidial suspension), or in a 24-well microtiter plate when the 24 h old biofilm was investigated (500 µL cells and 500 µL fungal suspension). An initial incubation of at least 4 h was necessary in order to facilitate the adhesion of the fungal conidia onto the A549 cells, as recommended in previously published studies [20]. The co-cultures were incubated together at 36 °C and 5% CO_2_. After the specific biofilm incubation time (4 h, 12 h or 24 h) was ended, the medium was discarded, the wells were washed gently (through pipetting) with PBS and the antifungal drugs (see Section 2.3) were added. After incubation for an additional 24 h, the biofilms were washed thrice with PBS to remove any non-adherent conidia. Briefly, 100 µL (4 h and 12 h) or 500 µL (24 h) of an XTT (0.5 mg/mL) (Santa Cruz Biotechnology, Dallas, TX, USA) plus 125 μM menadione (Sigma-Aldrich) solution was added to each well of the plates and incubated for 3 h at 36 °C and 5% CO_2_. Next, 80 µL of the supernatants were transferred from each well onto a fresh 96-well plate with a U-shaped bottom. The OD_492_ was measured using a microplate reader (Sunrise Tecan, Hombrechtikon, Switzerland). The workflow of the co-culture assay is summarized in Figure 1.

To rule out any cytotoxic effects through the antifungal compounds, a cytotoxicity assay (11465015001, Sigma-Aldrich) was performed (Appendix A, Figure A1). Briefly, A549 cells (1 × 10^5^ cells/mL) were incubated together with LLCZ, VCZ and AMB in 96-well flat-bottomed microtiter plates (100 µL cell suspension + 100 µL antifungal drug). The antifungal compounds were diluted in DMEM (12320032; Fisher Scientific, Gibco) to concentrations of 1, 2, 4 and 8 mg/L, followed by incubation at 36 °C and 5% CO_2_. A negative control was performed with DMEM and a toxicity control was performed with a 50% DMSO solution diluted in PBS. Upon incubation end, the medium was removed and the cells were incubated with XTT (1 mg/mL) plus 125 μM menadione for 3 h at 36 °C and 5% CO_2_. Then, 80 µL of the supernatants were transferred from each well onto a fresh 96-well plate with a U-shaped bottom. The OD_492_ was measured using a microplate reader (Sunrise Tecan, Hombrechtikon, Switzerland).

### 2.6. Confocal Laser Scanning Microscopy

The effect of the antifungal drugs was visualized by using confocal laser scanning microscopy (CLSM) (Zeiss, Oberkochen, Germany). For this purpose, the A549 cells were stained with a PKH26 cell membrane dye (Sigma-Aldrich, St. Louis, MO, USA).

#### 2.6.1. PKH26 Staining

The epithelial lung cells were stained as recommended by the manufacturer. Briefly, the cells were grown to a concentration of 2 × 10^6^ cells/mL and centrifuged into a loose pellet, followed by a re-suspension in the provided ethanolic dye and diluent. The reaction was stopped by adding a 1% bovine serum albumin solution. This was followed by two further washing steps in order to facilitate the complete removal of the unbound dye.

#### 2.6.2. Sample Preparation and Microscopy

The above-mentioned PKH26-stained A549 cells were incubated together with a fungal suspension (100 µL fungal suspension and 100 µL stained cells) in μ-slide 8-well glass-bottom plates (ibidi GmbH, Gräfelfing, Germany) at a concentration of 1 × 10^5^ cells/mL. Upon incubation completion, the co-culture was stained with calcofluor white to dye the fungal biofilm, followed by a fixation with methanol. The co-cultures could be visualized via CLSM using the Elyra LSM 710 instrument with lasers at 405 nm (UV laser diode—for calcofluor white), 543 nm (helium-neon laser—for PKH26) and a 20-fold magnification objective (Zeiss, Oberkochen, Germany) [9,20].

### 2.7. PCLS Preparation

The lungs were extracted from already-euthanized healthy mice with a C57BL/6 background, which were donated by the working group for Molecular Infection Immunology at the University Hospital Essen (working group Hansen). Briefly, the mouse lungs were filled with a 4% low-melting-point agarose solution (V2111; Promega, Madison, WI, USA) to maintain the rigidity of the lungs. The trachea was bound with a string and the lungs were cooled for at least 10 min in order for the agarose to solidify. The lungs were carefully excised from the mouse with scissors and stored in PBS at 4 °C. The lung lobes were then separately cut with a Leica—VT 1000S vibratome (Leica Biosystems, Wetzlar, Germany) into 400 µm thick tissue slices and stored in PBS at 4 °C. Upon use, the slices were placed on sterile round cover slips (Carl Roth, Karlsruhe, Germany) and transferred into 24-well microplates (Figure 2) [16,21].

### 2.8. Ex Vivo PCLS Infection and Treatment

Immediately after being placed in the 24-well microplates, the PCLS were infected with a 1 × 10^5^ cells/mL inoculum, which was applied directly on top of the slices. Treatment was either applied together with the inoculum (500 µL fugal suspension and 500 µL antifungal drug), both diluted in DMEM/F-12 medium (Fisher Scientific Waltham, Massachusetts, USA), or later on after an initial 24 h incubation of the biofilm at 36 °C and 5% CO_2_. At the end of incubation, the glass cover slips were transferred with sterile tweezers onto new 24-well microplates, where they were washed two times with PBS and incubated with 500 μL/well of a XTT solution (0.5 mg/mL) plus 25 μM menadione. The plates were incubated in the dark at 36 °C, the XTT supernatant (100 μL) was transferred from each well into a U-shaped 96-well plate, and the optical density was measured by using a plate reader (Sunrise Tecan, Hombrechtikon, Switzerland) at 492 nm. The effect of the antifungal compounds against biofilm formation on the PCLSs was assessed by comparing the OD_492_ of treated, untreated and uninfected slices (Figure 2).

### 2.9. Statistical Analysis

All experiments were performed at least three times (n = 3). Each isolate was pippeted in duplicates (two wells per treatment concentration) in order to calculate the metabolical activity percentage, and each treated isolate was compared to its untreated control. The statistical analysis was carried out using the program GraphPad Prism 9 (GraphPad Software Inc., La Jolla, CA, USA), by performing Dunnett’s multiple comparison test; significance was determined at *p* < 0.05 (*, *p* < 0.05; **, *p* < 0.01; ***, *p* < 0.001; and ****, *p* < 0.0001).

## 3. Results

In both assays, the anti-biofilm activity of each of the included antifungal agents was demonstrated. In the co-culture model, the strongest effect was noticed when the antifungal drugs were added to the infected epithelial lung cells shortly after incubation start (4 h). All three drugs showed a comparably strong effect of around 95% which could be detected via the XTT assay (Figure 3A). An independent cytotoxicity assay showed that the metabolic activity produced by the cells was not influenced by the antifungal drugs when they were added in the above-mentioned concentrations (Figure A1).

However, this effect significantly decreased with the maturation of biofilm after 12 h. While LLCZ and AMB both showed an inhibition of around 60%, the biofilm inhibitory effect for VCZ could be estimated at 40% (Figure 3B).

Treatment of the mature biofilm (24 h) was shown to be less effective, but significant inhibition was detected. The AMB-treated biofilm showed 84% of the biofilm metabolic activity while for LLCZ and VCZ treated mature biofilms a metabolic activity of 89% and 85% was recorded (Figure 3C).

The anti-biofilm effect of the antifungal compounds on A549 cells in early stages could also be illustrated via CLSM (Figure 4). The *A. fumigatus* biofilm grew unrestricted on the untreated A549 cells (Figure 4A), and the addition of VCZ, AMB and LLCZ (Figure 4B–D), after 4 h incubation time. The infected A549 cells treated once with the isolate-specific MIC were surrounded by damaged hyphae (stained in blue). No significant difference between treated and untreated co-cultures could be noted when treating the mature biofilm. For VCZ, similar results were noticed; however, more hyphal rests could be detected, in comparison to the LLCZ treatment. In the case of AMB, only conidial cells could be visualized, with no hyphal rests.

The ex vivo PCLS model showed slightly different results than the co-culture model. When the drug was directly added at incubation start (0 h), the largest decrease in activity was recorded for the infected PCLS treated with AMB (only 7% biofilm metabolic activity). Compared with AMB the effect of LLCZ and VCZ with 48% and 32% biofilm metabolic activity detected was lower (Figure 5A).

A weaker antifungal effect could again be seen when the drug was added at a later incubation time (24 h). VCZ showed the strongest inhibition with 71% of the biofilm metabolic activity detected on the PCLS compared to the non-treated control, followed by AMB (75%) and LLCZ (85%) (Figure 5B).

## 4. Discussion

The aim of this study was to explore the activity of traditional and novel antifungal compounds against *A. fumigatus* biofilm development in human-like models. Both assays were sufficient for testing the anti-biofilm activity against *Aspergillus*, whereas only the PCLS ex vivo model was capable of studying invasive mold growth conditions through providing tissue.

In diseases like IPA the fungus benefits from an aerobic and static environment, comparable to the environment found in commonly applied in vitro growth assays [13]. This static environment is optimal for the generation of biofilm [11]. We here applied a biofilm co-culture assay to show if living human cells had an impact on the biofilm development of *A. fumigatus*. We further investigated whether the different, human-like, growth conditions could affect the treatment efficiency.

The treatment with all three agents in the co-culture assay showed a significant inhibition of early-stage biofilm metabolic activity, with no obvious cytotoxic effects towards the human epithelial lung cells. AMB showed an overall consistent biofilm inhibition. VCZ and LLCZ, both azoles with a similar mechanism of action [22,23], showed a different metabolic activity inhibition, depending on the growth stage of the biofilm (Figure 3). LLCZ performed better against maturing biofilms (Figure 3B), while VCZ showed a slightly better inhibition of mature biofilms (Figure 3C). The effect of LLCZ showed no difference to the (invertebrate) in vitro anti-biofilm effects studied previously [10], suggesting that the human epithelial cells do not directly influence the LLCZ treatment of *A. fumigatus* biofilm.

The effect of all three antifungal agents investigated against biofilm in co-culture with the A549 cells was visualized by using CLSM (Figure 4). The fungistatic and fungicidal effects of the three compounds were in line with the ones described in the current literature [10,23,24,25].

However, despite the obvious advantages that the co-culture model presents, it has several drawbacks. Two-dimensional cell culture models rely on immortalized cell lines, which have altered metabolisms and gene expression profiles [26]. While the human epithelial lung cells (A549) may simulate some of the conditions present in in vivo models, they are still a model that cannot entirely replicate the cellular richness and spatial complexity of the lung [15].

PCLS, in contrast to the epithelial cells used in the co-culture assay, retain the most important aspects of the lung: the complex architecture, the mechanical aspects and the specific lung cells [27,28]. Here, AMB showed the most promising effect of the tested compounds overall, while LLCZ showed a significant decrease in activity in the PCLS model compared to the co-culture model. An explanation for the difference in the activity of LLCZ could be that the retention of the drug in tissue is significantly poorer than when it is applied topically [29]. It is possible that upon infection, the *A. fumigatus* hyphae infiltrated the alveolar structures found in the PCLS and built biofilm structures that were harder to reach by LLCZ, due to its lower tissue penetration capability. The drug is a weak base, which means it is not easily soluble. Thus, it is easily retained in the skin and cannot penetrate the tissue to reach deeper-seated infections [29]. Due to a lack of proper imaging techniques of the PCLS, this could not be confirmed.

In contrast to the azoles, AMB was capable of biofilm disruption. A comparison of AMB activity in the two distinct models showed a greater metabolic activity inhibition in the PCLS ex vivo model (93%) than in the co-culture in vitro model (71%). AMB is a fungicide characterized through prolonged residence times in tissue, explaining its strong effect in the PCLS model [30]. Additionally, this 22% difference in activity could be explained with the differences in the two experimental setups. While the biofilms of the co-culture model were washed after 4 h of incubation and then treated, the PCLS were treated directly at incubation start.

The effect of all three compounds was significantly reduced when applied to mature biofilm, as previously described in the literature [10,11]. VCZ and AMB showed an overall good activity, most likely indicating that the planktonic growth of the fungus was inhibited before it was able to develop a protective biofilm structure.

Additionally, the activity of antimicrobials varies in the different stages and growth conditions of the pathogens. For example, the MICs of antifungals are higher for biofilm-associated cells in comparison to those of planktonic origin. One difference could result in the composition of the ECM [31]. The ECM is not present at all in vitro under liquid shaking conditions. However, it is present in static in vitro growth conditions, acting as cohesive linkage for the hyphae, most likely due to the presence of extracellular DNA [31]. A similar situation can be found in vivo, in IPA patients, where the ECM is produced on the surface of the hyphae [32]. It is also important to consider that in humans, fungal biofilms may develop differently depending on the types of patients that are being treated [14]. Different pathologies (such as aspergilloma or IPA) may show a different ECM composition [14]. For this reason, we believe that the use of human-like models for the study of fungal biofilm models in pre-clinical studies can be helpful.

Despite the benefits of the above-shown human-like models used in this study, certain limitations must also be addressed. Firstly, the concentration difference between LLCZ and the other two antifungals needs to be addressed. Future experiments would need to include a wide antifungal drug concentration range to clarify the association between drug activity and concentration and optimize any future therapeutic concentrations. While it is speculated that the ECM is most likely responsible for the decrease in the efficiency of the antifungal drugs against mature biofilm, the present study does not make a direct correlation between the two. For this reason, future experiments will be supplemented with immunocytochemical (in vitro) or immunohistochemical (ex vivo) data for the direct detection of ECM polysaccharides such as galactomannan, galactosaminogalactan or α-1,3 glucan. Biofilm staining experiments (e.g., with crystal violet) are also considered. Furthermore, no method for assessing the viability of the PCLS was used in this study. For this reason, future experiments will include an in-depth histopathological analysis, to ensure the model’s stability during the experimental period.

## 5. Conclusions

In conclusion, the results of this study indicate the benefits of using human-like models in laboratory settings for the study of anti-biofilm activity. While both assays are suitable for testing anti-biofilm activity in fungi, the PCLS model has advantages over the co-culture model, including a tissue for invasive mold growth. Both models complement the already-established methods used for biofilm assessment and the results from this study could contribute towards a possible optimization of future fungal biofilm experiments in murine models.

## Figures and Tables

**Figure 1 microorganisms-13-02040-f001:**
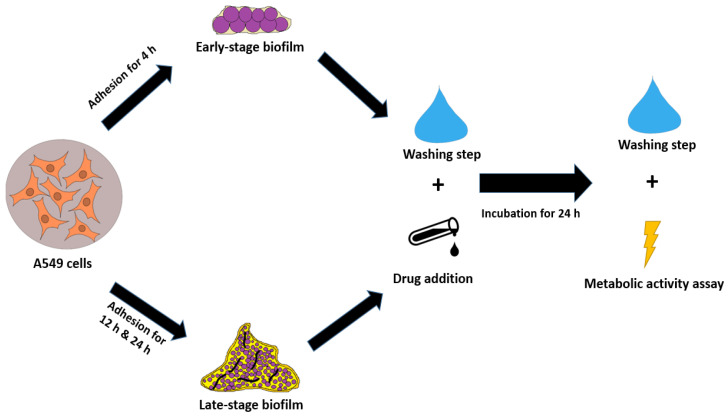
A549/biofilm co-culture assay for immature (4 h), maturing (12 h) and mature (24 h) biofilm.

**Figure 2 microorganisms-13-02040-f002:**
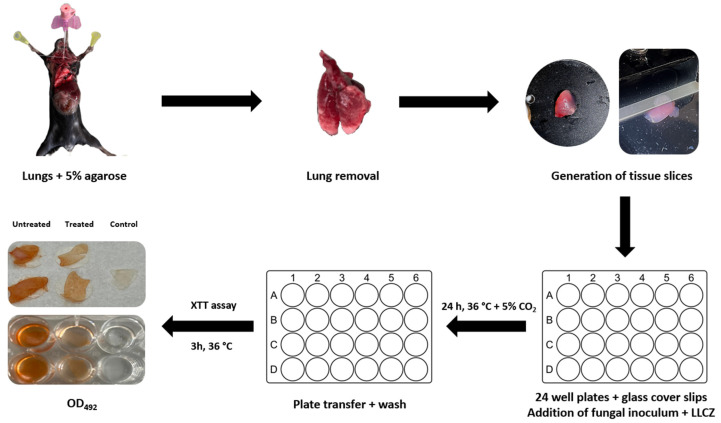
Generation of precision cut lung slices and assessment of biofilm metabolic activity (early stage).

**Figure 3 microorganisms-13-02040-f003:**
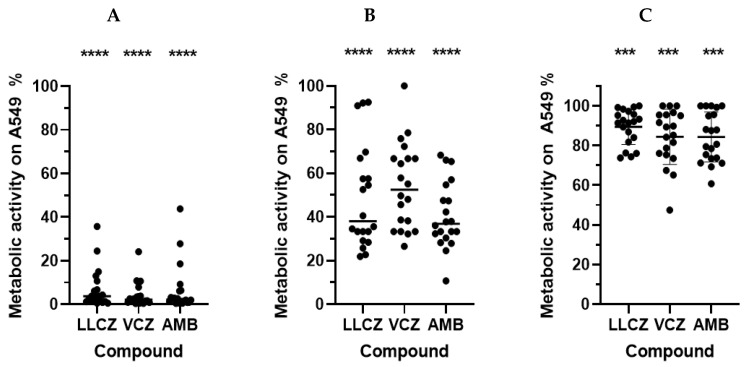
XTT assay determined the metabolic activity of *A. fumigatus* biofilm, in co-culture with A549 epithelial lung cells, when treated with luliconazole (LLCZ), voriconazole (VCZ) and amphotericin B (AMB) (N = 20 isolates). Isolates were treated with the minimum inhibitory concentration (for LLCZ) and 1 mg/L (which corresponds to the EUCAST breakpoints of VCZ and AMB). The antifungal drugs were added after an initial incubation time (36 °C) of 4 h (**A**) 12 h (**B**) or 24 h (**C**) followed by a subsequent 24 h incubation at 36 °C. The isolates were then washed twice with 1 × PBS and incubated with XTT (0.5 mg/mL) and menadione (25 µM) for 3 h. The optical density was measured at 492 nm (***, *p* < 0.001; and ****, *p* < 0.0001).

**Figure 4 microorganisms-13-02040-f004:**
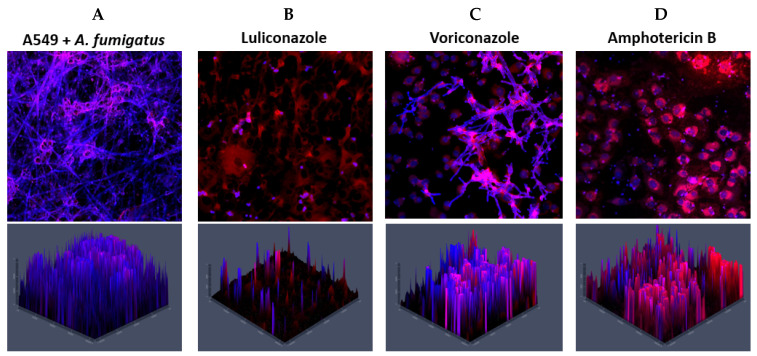
The membrane of the A549 cells (red) was stained with PKH26 while the *A. fumigatus* biofilm (blue) was stained with calcofluor white. The images show CLSM renderings (2D and 2.5D) of *A. fumigatus* biofilm/A549-cell co-cultures untreated (**A**) and treated with 0.002 mg/L luliconazole (LLCZ) (**B**), 1 mg/L voriconazole (VCZ) 1 (**C**) and mg/L amphotericin B (AMB) (**D**) after 4 h of incubation.

**Figure 5 microorganisms-13-02040-f005:**
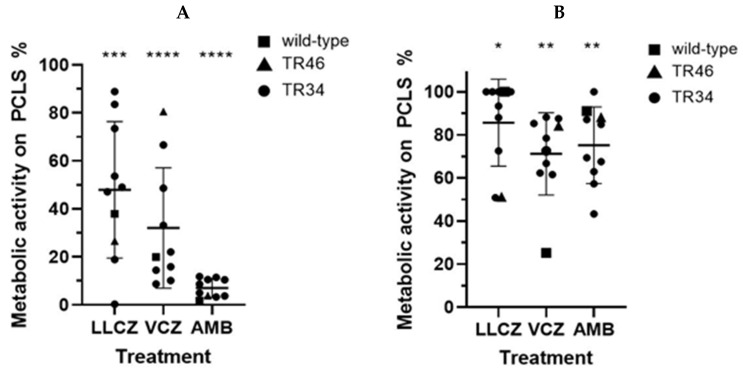
The XTT assay determined the metabolic activity of *A. fumigatus* biofilm formed on the precision cut lung slices (PCLS), when treated with luliconazole (LLCZ), voriconazole (VCZ) and amphotericin B (N = 10 isolates). Isolates were treated with the minimum inhibitory concentration (for LLCZ) and 1 mg/L (which corresponds to the EUCAST breakpoints of VCZ and AMB). The antifungal drugs were added either at the start of incubation (**A**) or 24 h after the initial incubation (**B**) followed by a subsequent 24 h incubation at 36 °C. The isolates were then washed twice with 1 × PBS and incubated with XTT (0.5 mg/mL) and menadione (25 µM) for 3 h. The optical density was measured at 492 nm (*, *p* < 0.05; **, *p* < 0.01; ***, *p* < 0.001; and ****, *p* < 0.0001).

**Table 1 microorganisms-13-02040-t001:** A list of all the *A. fumigatus* isolates used in this study.

Isolate	Source	Mutations
1950	BS	TR34/L98H
1954 *	TS	TR34/L98H
1959	TS	TR34/L98H
1962	BAL	TR34/L98H
1966	TS	TR34/L98H
1971	Spt	TR34/L98H
1972	BS	No mutation
1978	BS	TR34/L98H
1986	Spt	TR34/L98H
2047	BAL	TR34/L98H
2107 *	BAL	TR46
2119 *	Spt	TR34/L98H
2133	Spt	TR34/L98H
2135 *	Spt CF	TR34/L98H
2514 *	BS CF	TR34/L98H
2515 *	Spt CF	TR34/L98H
2607 *	Spt CF	TR34/L98H
2608 *	Spt CF	TR34/L98H
ATCC 204305 *	ATCC	Wild-type
m1215 *	Spt CF	No mutation

TS. tracheal secretion; BAL. broncho-alveolar lavage; ATCC. American Type Culture Collection; BS. bronchial secretion; Spt. sputum; CF. cystic fibrosis patient; * strains used in the PCLS assay.

## Data Availability

The original contributions presented in this study are included in the article. Further inquiries can be directed to the corresponding author.

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
