# Peer review of "Human-like Biofilm Models to Study the Activity of Antifungals Against Aspergillus fumigatus"

_microorganisms, 2025, doi:10.3390/microorganisms13092040_

Round 1
Reviewer 1 Report
Comments and Suggestions for Authors
Good article, it is recommended to publish after revised.
1. In table 1. It is suggested to add a column to explain the types of mutations of each isolates and delete the third column.
2. Add statistical method in the method section.
3. Figure A1 should be Figure S1, Is there biological replication in this experiment?
Author Response
- In table 1. It is suggested to add a column to explain the types of mutations of each isolates and delete the third column.
Reply 1: The third column was replaced with a column that describes the isolate mutations
- Add statistical method in the method section.
Reply 2: A “Statistical methods” paragraph has been added to the methods section
- Figure A1 should be Figure S1, Is there biological replication in this experiment?
Reply 3: The title of the figure has been changed. There is no biological replication in the experiment, due to the use of only one cell line. We added this detail in the figure legend.
Reviewer 2 Report
Comments and Suggestions for Authors
Dear authors;
I would like to suggest that the introduction provide a more comprehensive overview of the fungal species (A. fumigatus), including its biology, taxonomic classification, macro and micro morphology, and mode of infection in the human host.
Furthermore, I suggest that the conditions under which the inoculum was prepared be better specified in the inoculum preparation section (lines 76-82). This was not clear from the way it was written.
Best regards.
----
Author Response
- I would like to suggest that the introduction provide a more comprehensive overview of the fungal species ( fumigatus), including its biology, taxonomic classification, macro and micro morphology, and mode of infection in the human host.
Reply 1: We agree that the introduction can use extra depth therefore, a detailed paragraph regarding A. fumigatus has been added to the introduction.
- Furthermore, I suggest that the conditions under which the inoculum was prepared be better specified in the inoculum preparation section (lines 76-82). This was not clear from the way it was written.
Reply 2: We agree that the paragraph describing the methodology could be improved and a more detailed version has been added.
Reviewer 3 Report
Comments and Suggestions for Authors
This article focuses on the study of antifungal drug activity against biofilms of Aspergillus fumigatus. By establishing co-cultured A549 cell models and PCLS precision lung slice models, the anti-biofilm effects of voriconazole, amphotericin B, and luliconazole were systematically evaluated.
- The introduction section can be supplemented with the recent application progress of human-like models (such as PCLS) in fungal biofilm research, in order to enhance the innovativeness of the model selection in this study.
2.In section 2.3, it is mentioned that voriconazole and amphotericin B were diluted to 1 mg/L, and luliconazole was used at strain-specific MICs. Is it necessary to set other concentration gradients to clarify the association between drug activity and concentration and determine the optimal therapeutic concentration?
- In the co-culture model described in the article, the inhibition rate of LLCZ at 12 hours is 60%, which is higher than that of VCZ at 40%. However, at 24 hours, VCZ slightly outperforms LLCZ. Is this difference related to changes in the extracellular matrix (ECM) components of the biofilm? It is recommended to supplement with the detection of biofilm structure for further explanation.
4.In Figure 3, the XTT assay results are presented through the chart. Is it necessary to specify data such as the number of independent experiments and sample size in the figure legend?
- Insufficient Mechanistic Discussion: The study only focused on the effect of drugs on the metabolic activity of biofilms, without in-depth analysis of the mechanism of drugs on biofilm structure (such as extracellular matrix components) and fungal virulence factors, which limits the in-depth interpretation of the results.
- The survival time of PCLS and histopathological changes after infection were not clarified, making it impossible to confirm the stability of the model during the experimental period.
Author Response
1. The introduction section can be supplemented with the recent application progress of human-like models (such as PCLS) in fungal biofilm research, in order to enhance the innovativeness of the model selection in this study.
Reply 1: We agree with the reviewer and added a paragraph in the introduction, outlining the fact that this model has been previously used for the study of bacteria, however not for fungi.
2. In section 2.3, it is mentioned that voriconazole and amphotericin B were diluted to 1 mg/L, and luliconazole was used at strain-specific MICs. Is it necessary to set other concentration gradients to clarify the association between drug activity and concentration and determine the optimal therapeutic concentration?
Reply 2: We thank the reviewer for this remark and have added this as a recommendation in lines 376-379.
3. In the co-culture model described in the article, the inhibition rate of LLCZ at 12 hours is 60%, which is higher than that of VCZ at 40%. However, at 24 hours, VCZ slightly outperforms LLCZ. Is this difference related to changes in the extracellular matrix (ECM) components of the biofilm? It is recommended to supplement with the detection of biofilm structure for further explanation.
Reply 3: We agree and added a recommendation for future experiments in lines 385-388, where we also describe further limitations.
4. In Figure 3, the XTT assay results are presented through the chart. Is it necessary to specify data such as the number of independent experiments and sample size in the figure legend?
Reply 4: We agree that some of the information in the legend was redundant. The sample size is required to avoid any confusion. The two experiments utilize a different number of A. fumigatus isolates (N= 20 vs 10). Other redundant statistical data has been moved to the methods section.
5. Insufficient Mechanistic Discussion: The study only focused on the effect of drugs on the metabolic activity of biofilms, without in-depth analysis of the mechanism of drugs on biofilm structure (such as extracellular matrix components) and fungal virulence factors, which limits the in-depth interpretation of the results.
Reply 5: We agree with the reviewer that this study has multiple limitations, such as those listed above. We added an “ECM component screening” as a recommendation in lines 380-385.
6. The survival time of PCLS and histopathological changes after infection were not clarified, making it impossible to confirm the stability of the model during the experimental period.
Reply 6: Histopathological analysis of the tissue is, indeed, vital for the future use of this model in our experiments. Due to constraints, we could not perform it. Even though the tissue was freshly prepared prior to each experiment and stored at 4 °C, we have no concrete evidence on how the tissue changed during the experiment. We have added this as a limitation in lines 385-388 and we thank the reviewer for pointing it out.
Reviewer 4 Report
Comments and Suggestions for Authors
Major point. The identity of fungal isolates is unclear.
Was DNA sequencing of some barcoding locus performed? The Authors mention that A. fumigatus strains were previously identified via internal transcribed spacer 1 sequencing and refer to the publication Ref. 12 (Lines 62-63). Ref. 12 has nothing to do with the described A. fumigatus strains. Its authors performed DNA sequencing of the whole ITS region, including ITS1, 5.8S rRNA-coding gene and ITS2 (the primers ITS1 and ITS4). Another referenced publication by the Authors described the employed isolates (Ref. 8). It also mentions DNA sequencing performed earlier by Seufert et al., PMID 29684150. However, DNA sequencing in the latter publication was limited to the cyp51A gene. In Aspergillus, the taxonomic resolution of ITS region is limited, implying a possibility of species misidentification [1]. Therefore, the identification should be done either with BenA and CaM sequences [2, 3] or whole genome sequences [4], as described in the mentioned papers. In both cases, the obtained sequences must be made publicly available prior to publication.
The search with the keywords "human-like model" "biofilm" in Google Scholar retrieved 16 results. With the word “fungus” added, it retrieved zero results. As the paper is not the first one of its kind, I suggest revising the list of keywords.
[1] Houbraken J, Visagie CM, Frisvad JC. Recommendations To Prevent Taxonomic Misidentification of Genome-Sequenced Fungal Strains. Microbiol Resour Announc. 2021 Dec 2;10(48):e0107420. doi: 10.1128/MRA.01074-20. Epub 2021 Dec 2. PMID: 34854710; PMCID: PMC8638587.
[2] Hubka V, Barrs V, Dudová Z, SklenáÅ™ F, Kubátová A, Matsuzawa T, Yaguchi T, Horie Y, Nováková A, Frisvad JC, Talbot JJ, KolaÅ™ík M. Unravelling species boundaries in the Aspergillus viridinutans complex (section Fumigati): opportunistic human and animal pathogens capable of interspecific hybridization. Persoonia. 2018 Dec;41:142-174. doi: 10.3767/persoonia.2018.41.08. Epub 2018 Jun 21. PMID: 30728603; PMCID: PMC6344812.
[3] Houbraken J, Kocsubé S, Visagie CM, Yilmaz N, Wang XC, Meijer M, Kraak B, Hubka V, Bensch K, Samson RA, Frisvad JC. Classification of Aspergillus, Penicillium, Talaromyces and related genera (Eurotiales): An overview of families, genera, subgenera, sections, series and species. Stud Mycol. 2020 Jun 27;95:5-169. doi: 10.1016/j.simyco.2020.05.002. PMID: 32855739; PMCID: PMC7426331.
[3] Steenwyk JL, Balamurugan C, Raja HA, Gonçalves C, Li N, Martin F, Berman J, Oberlies NH, Gibbons JG, Goldman GH, Geiser DM, Houbraken J, Hibbett DS, Rokas A. Phylogenomics reveals extensive misidentification of fungal strains from the genus Aspergillus. Microbiol Spectr. 2024 Apr 2;12(4):e0398023. doi: 10.1128/spectrum.03980-23. Epub 2024 Mar 6. PMID: 38445873; PMCID: PMC10986620.
Author Response
1. Major point. The identity of fungal isolates is unclear.
Was DNA sequencing of some barcoding locus performed? The Authors mention that A. fumigatus strains were previously identified via internal transcribed spacer 1 sequencing and refer to the publication Ref. 12 (Lines 62-63). Ref. 12 has nothing to do with the described A. fumigatus strains. Its authors performed DNA sequencing of the whole ITS region, including ITS1, 5.8S rRNA-coding gene and ITS2 (the primers ITS1 and ITS4). Another referenced publication by the Authors described the employed isolates (Ref. 8). It also mentions DNA sequencing performed earlier by Seufert et al., PMID 29684150. However, DNA sequencing in the latter publication was limited to the cyp51A gene. In Aspergillus, the taxonomic resolution of ITS region is limited, implying a possibility of species misidentification [1]. Therefore, the identification should be done either with BenA and CaM sequences [2, 3] or whole genome sequences [4], as described in the mentioned papers. In both cases, the obtained sequences must be made publicly available prior to publication.
Reply 1: We thank the remark for the suggestion; it would be very useful to perform identification using the BenA and CaM sequences. We usually identify A. fumigatus isolates from clinical samples macro- and microscopically. Additionally, the AsperGenius real-time PCR from Pathonostics is used for species identification and determination of underlying mutation. This PCR is commercially available for Species identification and resistance analysis. Isolates collected before 2016 were analysed in a nested PCR.
Additionally, the routine diagnosis in the lab benefits from external quality controls as required by the ECMM standard and ring trials performed multiple times a year.
2. The search with the keywords "human-like model" "biofilm" in Google Scholar retrieved 16 results. With the word “fungus” added, it retrieved zero results. As the paper is not the first one of its kind, I suggest revising the list of keywords.
Reply 2: We thank the reviewer for their remark and have updated the list of keywords.
Reviewer 5 Report
Comments and Suggestions for Authors
In this manuscript, the authors study the effect of two established drugs for invasive fungal (mold)infections, amphotericin and voriconazole, as well as an agent used more commonly as a topical agent, luliconazole, on planktonic or biofilm-embedded fungal cells. Fungal cells were incubated either on a cell line (A549) or on lung slices. Regarding the experiments on A549 cells, fungal cells were co-incubated with the cells for 4, 12 and 24 hrs, before the addition of antifungals, whereas regarding the experiments with mice's lung slices, antifungals were added either simultaneously with fungal cells or after 24 h. The authors conclude that biofilm embedded cells are far more resistant to antifungals in the experiments using the cell line. Amphotericin had the best activity when added simultaneously with the fungus in experiments using lung slices, whereas voriconazole had the best effect, but close to amphotericin, when added 24 h later at the same experiment. Finally, the authors conclude that use of lung slices is superior to cell line. In lung slices experiments, the inferiority of luliconazole is revealed and is attributed to poorer tissue penetration capability.
The manuscript is well written. The experiments are well organized and described. Conclusions are supported from results.
Minor points:
Line 204: "the biofilm inhibitory effect could be estimated at 40 %" , do the authors refer to the effect of voriconazole?, please add the information.
Line 345-6: "the antifungal drug susceptibilities of biofilm-associated cells are higher in comparison to those of planktonic cells". Maybe the authors mean "the minimum inhibitory concentrations MICs of antifungals are higher for biofilm-ssociated cells in comparison to those of planktonic cells".
Author Response
- Line 204: "the biofilm inhibitory effect could be estimated at 40 %" , do the authors refer to the effect of voriconazole?, please add the information.
- Line 345-6: "the antifungal drug susceptibilities of biofilm-associated cells are higher in comparison to those of planktonic cells". Maybe the authors mean "the minimum inhibitory concentrations MICs of antifungals are higher for biofilm-ssociated cells in comparison to those of planktonic cells".
Reply 1+2: We thank the reviewer for pointing out the two mistakes in our manuscript. We have adapted the highlighted paragraphs accordingly.
Round 2
Reviewer 3 Report
Comments and Suggestions for Authors
none
Reviewer 4 Report
Comments and Suggestions for Authors
I have no further questions.